# DArTSeq SNP-based markers revealed high genetic diversity and structured population in Ethiopian cowpea [Vigna unguiculata (L.) Walp] germplasms

Selamawit Ketema[1,2], Bizuayehu Tesfaye[2], Gemechu Keneni[3], Berhanu Amsalu Fenta[1], Ermias Assefa[4], Nicolas Greliche[5], Eunice Machuka[6], Nasser Yao[6]*

1 Ethiopian Institute of Agricultural Research, Melkassa Research Center, Melkassa, Ethiopia, 2 School of Plant and Horticultural Science, Hawassa University, Hawassa, Ethiopia, 3 Ethiopian Institute of Agricultural Research, Addis Ababa, Ethiopia, 4 Ethiopian Biotechnology Institute, Genomic Research Directorate, Addis Ababa, Ethiopian, 5 Statistics for Sustainable Development, Reading, United Kingodm, 6 Biosciences Eastern and Central Africa - International Livestock Research Institute (BecA - ILRI) Hub, Nairobi, Kenya

* nasseryao@gmail.com

**Data Availability Statement:** All relevant data are within the manuscript and its Supporting Information files.

## Abstract

Cowpea [Vigna unguiculata (L.) Walp] is one of the important climate-resilient legume crops for food and nutrition security in sub-Saharan Africa. Ethiopia is believed to harbor high cowpea genetic diversity, but this has not yet been efficiently characterized and exploited in breeding. The objective of this study was to evaluate the extent and pattern of genetic diversity in 357 cowpea accessions comprising landraces (87%), breeding lines (11%) and released varieties (2%), using single nucleotide polymorphism markers. The overall gene diversity and heterozygosity were 0.28 and 0.12, respectively. The genetic diversity indices indicated substantial diversity in Ethiopian cowpea landraces. Analysis of molecular variance showed that most of the variation was within in the population (46%) and 44% between individuals, with only 10% of the variation being among populations. Model-based ancestry analysis, the phylogenetic tree, discriminant analysis of principal components and principal coordinate analysis classified the 357 genotypes into three well-differentiated genetic populations. Genotypes from the same region grouped into different clusters, while others from different regions fell into the same cluster. This indicates that differences in regions of origin may not be the main driver determining the genetic diversity in cowpea in Ethiopia. Therefore, differences in sources of origin, as currently distributed in Ethiopia, should not necessarily be used as indices of genetic diversity. Choice of parental lines should rather be based on a systematic assessment of genetic diversity in a specific population. The study also suggested 94 accessions as core collection which retained 100% of the genetic diversity from the entire collection. This core set represents 26% of the entire collection pinpointing a wide distribution of the diversity within the ethiopian landraces. The outcome of this study provided new insights into the genetic diversity and population structure in Ethiopian cowpea genetic resources for designing effective collection and conservation strategies for efficient utilization in breeding.

**Funding:** The laboratory work was supported by the BecA-ILRI Hub through the Africa Biosciences Challenge Fund (ABCF) program. The ABCF Program is funded by the Australian Department for Foreign Affairs and Trade (DFAT) through the BecA-CSIRO partnership; the Syngenta Foundation for Sustainable Agriculture (SFSA); the Bill & Melinda Gates Foundation (BMGF); the UK Department for International Development (DFID) and; the Swedish International Development Cooperation Agency (Sida). The authors are also grateful to the Collaborative Crop Research Project of McKnight Foundation for the financial support for the collection of the landraces and maintenance of the germplasm. The funders had no role in study design, data collection and analysis, decision to publish, or preparation of the manuscript.

**Competing interests:** The authors have declared that no competing interests exist.

## Introduction

Cowpea [*Vigna unguiculata (L.) Walp.*, 2n = 2x = 22] originated and was domesticated in Africa though the exact location of origin of domestication is still a matter of speculation and different authors suggest different areas in Africa, Northeastern Africa including Ethiopia [1–4], Central Africa [5], Southern Africa [6], and West Africa [5,7,8]. There are five known subspecies of cowpea, of which three are cultivated (*unguiculata*, *cylindrical* and *sesquipedalis*) and two are wild (*dekindtiana* and *mensensis*) [9,10]. In Ethiopia, all five subspecies are known to exist, and are of particular significance, being landraces of subspecies *unguiculata* and *cylindrical*, particularly in the drought-prone areas of eastern Ethiopia [11,12]. These subspecies are also grown in the northern, southwestern and southern parts of Ethiopia [12]. Thulin (1989) [13] reported that the subspecies *sesquipedalis* and *dekindtiana* are also cultivated in northern Ethiopia [14].

Cowpea is an important legume crop estimated to be grown on more than 11 million hectares of land, with an annual worldwide production of over 6 million tons, of which 96% is grown in Africa [15]. Cowpea is grown for different purposes, mainly as a source of staple food and nutritional security for farmers in sub-Saharan Africa. Cowpea is also useful for sustaining the farming system in Africa through its ability to fix atmospheric nitrogen and its tolerance to a wide number of abiotic stresses, including drought, heat, low soil pH and soil nutrient deficiency stress [16]. Cowpea, like other legumes, plays an important role when used in rotation with cereals by breaking the life cycles of pathogens of cereals [17,18]. In terms of nutrition security, cowpea is an affordable source of carbohydrate, protein, essential minerals, vitamins and folates, particularly to poor people who cannot afford animal-based diets [19,20].

Conventional cowpea breeding, particularly in African countries such as Nigeria, Senegal, Uganda and Tanzania, dates back to the 1960s [21]. Evidence shows that despite its many merits, cowpea breeding has suffered at least four main challenges. Firstly, the cowpea gene pool may be narrow, due partly to a genetic bottleneck during domestication. Secondly, genetic variation may be restricted by the 'founder effects' and limited germplasm exchange [22]. Thirdly, the crop has been awarded low priority and remained rather more orphaned in a number of countries, such as Ethiopia, where only limited efforts have been made to improve its productivity and utilization [23]. Fourthly, national cowpea improvement efforts in countries such as Ethiopia relied excessively on exotic genetic materials, particularly those from the International Institute of Tropical Agriculture (IITA). While broadening the genetic basis of breeding materials through incorporation of the exotic gene pool itself is the right way, landraces also have considerable breeding value, particularly under marginal conditions, as they contain valuable adaptive genes to different circumstances [24,25].

Knowledge of the extent and pattern of genetic diversity in a given gene pool provides plant breeders with an opportunity to develop new varieties with desirable traits [26]. Firstly, genetic gain from direct selection depends on the magnitude of genetic variability among the germplasms, heritability of a given trait in a given environment and the level of selection intensity applied [27,28]. Secondly, it is believed that crosses between genetically diverse parents are likely to produce higher heterosis, desirable genetic recombination and segregation in their progeny [29]. Apart from yield trials that have been conducted mostly on exotic genetic materials in Ethiopia, scientific evidence on the extent and pattern of genetic diversity in local cowpea landraces is limited. Among the few studies conducted, Belayneh *et al.* [30] assessed the genetic diversity in Ethiopian landraces using simple sequence repeats (SSR) markers and detected three-well differentiated ancestral populations. Genetic resource management, including building core collection for efficient space monitoring, is one of the common practice used worldwide to drive germplasm enhancement for future breeding. Core collections

are subsamples of larger genetic resources collections which are created in order to include a minimum number of accessions representing the maximum if not the whole diversity of the original collection [31].

Molecular markers have been utilized in several crop species to ascertain the existence of an adequate amount of genetic diversity in a given gene pool. Over the last few decades, several marker technologies have been developed for cowpea, starting from the early days of the iso-zyme [32,33] to the relatively recent time of randomly amplified DNA fingerprinting and random amplified polymorphic DNA [34–38], amplified fragment length polymorphism [3,22], randomly amplified microsatellite fingerprinting and microsatellite [16,30,39–43] markers. With the more recent developments in molecular genetics, however, the single nucleotide polymorphism (SNP) method has emerged as a more precise, cost-effective and faster method that offered a lot of comparative advantages to the aforementioned molecular markers [44]. SNP markers are also common and are found throughout the genome; they are stable and readily assayed using high-throughput genotyping protocols with automated data analysis. Although SNP markers can be observed through various experimental protocols, at present, genotype-by-sequencing (GBS) is the most popular approach to their identification in plants [45].

GBS technologies produced robust marker genotypes and tens to thousands of them, in contrast to previous SNP arrays [46]. GBS has been used to build core collection as well as investigating the genetic diversity and population structure of many crop species, including cowpea [47,48]. The objectives of this study were, therefore, to determine the extent of genetic diversity comprehensively, to estimate the levels of population structure in Ethiopian cowpea germplasm collections using high-throughput GBS-derived SNP markers and to identify the minimum number of accessions capturing the maximum diversity for conservation.

## Materials and methods

### Plant materials

Three hundred and sixty-one cowpea genotypes were used for this study, of which 314 landraces collected from different regions in Ethiopia were kindly provided by Melkasssa Agricultural Research Center of the Ethiopian Institute of Agricultural Research. The original collection areas of the cowpea landraces in Ethiopia are given in Fig 1, along with a description of all sets of the test genotypes provided in S1 Table. Fig 1 was constructed using the software DIVA-GIS as described by Hijmans et al. (2012) [49] using the GPS coordinates of the collection sites (S1 File). The 314 collected cowpea landraces comprised 70 genotypes collected from Amhara, 94 from Southern Nations, Nationalities and Peoples Regional State (SNNPRS), 59 from Gambella, 49 from Oromia and 42 from Tigray. About 40 breeding lines previously introduced from the IITA and six released varieties were also included in the study and considered improved varieties. All sets of the cowpea test materials used in this study, hereafter called genotypes, were regarded as a population and each grouping, based on geographic regions and breeding status (landraces and improved varieties), was regarded as a subpopulation.

### DNA extraction and sequencing

The genotypes were grown in a seed germination chamber (Conviron) at the Biosciences eastern and central Africa-International Livestock Research Institute (BecA-ILRI) hub using cell trays. Three seeds of each genotype were sown per tray. Ten-day-old leaf material was collected from the three seedlings and the pooled leaf samples were frozen in liquid nitrogen and stored at −80˚C for later use. Genomic DNA (gDNA) was extract from the frozen tissue according to the CTAB protocol, with some modifications [50]. The quantity of extracted DNA was

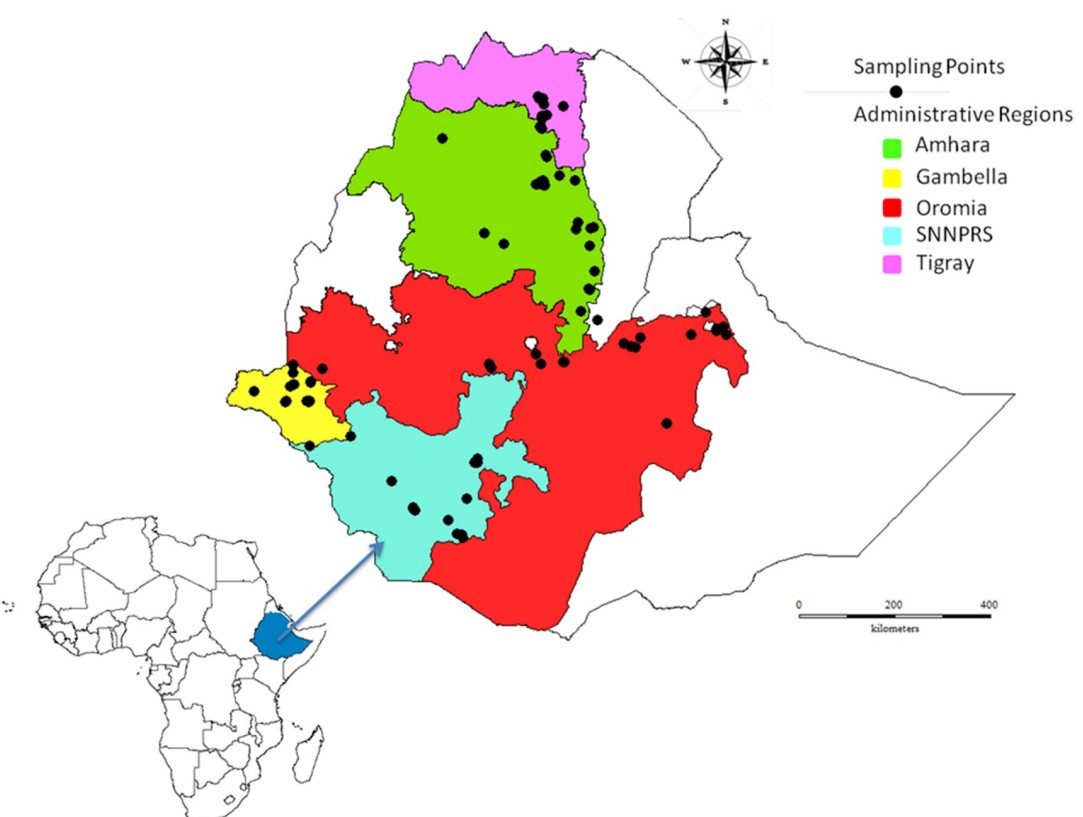

**Fig 1. Map of Ethiopia showing the collection sites for the different cowpea landraces from different eco-geographical regions.** The map was constructed using the DIVA-GIS software as described by Hijmans et al., (2012) [49].

checked using a Thermo Scientific NanoDrop Spectrophotometer 2000c. The quality of the DNA was confirmed on 0.8% agarose gel run in 1% TAE buffer at 70 V for 45 minutes. After the quality had been checked, 40 μl of a 50 ng/μl gDNA of each sample of 359 cowpea genotypes was sent for whole genome scanning using Genotyping by sequencing technology as described by Elshire et al. (2011) [45], using DArTseqTM technology (https://www. diversityarrays.com/) of the Integrated Genotype Service and Support platform in Nairobi, Kenya. GBS was performed by using a combination of DArT complexity reduction methods and next generation sequencing following protocols described in [51–53]. The complexity reduction method used involves digestion with the methylation-sensitive restriction enzyme, PstI. In conjunction with digestion using this relatively rarely-cutting restriction enzyme (six bp recognition site plus methylation sensitivity; Gruenbaum *et al*. 1981 [54]), an enzyme with frequent cutting capabilities. In this study, the frequently-cutting enzymes AluI, BstNI, TaqI or MseI were used. PCR adapters were ligated to the PstI fragment ends, and the PCR-amplification was performed using primers complementary to the PstI adapters. Only those fragments with PstI adapters at both ends were amplified.

## SNPs calling and data filtering

The data were previously trimmed following the DArTSeqTM technology based on the following filter criteria; markers/SNPs with call rate > 97% and allele-calling equal or greater than 98% were selected. Genotypes with read depth less than the threshold were coded as missing. SNP markers with high proportion of missing data (>10%), individuals who have high rates of

genotype missingness (>10%) and rare SNPs with <5% minor allele frequency (MAF) were discarded for further analysis using R software (version 2.8.3). The most informative SNPs were selected based on a threshold PIC value equal or higher than 0.2. Finally 357 cowpea genotypes and 6,498 (32%) of SNP markers were maintained for further analysis (S2 File).

## Genetic diversity and phylogenic analysis

The population's genetic structure was analyzed by conducting an analysis of molecular variance [55] using poppr package in the R version 2.8.3 [56,57]. The phylogenetic relationships of the subpopulations were generated based on pair-wise fixation indexes using the StAMPP package [58] and neighbor joining trees were constructed using the dartR package in R.

Pair-wise genetic frequency-based dissimilarity or distance matrix between individuals was calculated according to Euclidean distance as implemented in the R environment. The resulting dissimilarity matrix was subjected to tree construction using the unweighted pair group method analysis (UPGMA) employing the same software with the ggdendro and ggplot2 packages. Phylogenetic trees were constructed in R implementing the hclust algorithm, with the UPGMA relevant agglomeration method.

## Population structure analysis

To infer the population structure of Ethiopian cowpea landraces, three complementary methods were used: 1) Bayesian model-based clustering algorism (STRUCTURE software) [59], 2) discriminant analysis of principal components (DAPC) and 3) principal coordinate analysis (PCoA). The structure analysis was run five times for each K value (K = 1 to 10) using a burn-in period of 50 000 with 100 000 Markov Chain Monte Carlo iterations, assuming an admixture model and uncorrelated allele frequencies. The most probable value of K for each test was detected by ΔK [60], using the web-based program Structure Harvester [61]. CLUMPP v.1.1.2 [62] was used to align cluster assignment from independent runs using the in-files generated by structure Harvest. Bar plots were generated with average results of runs for the most probable K value, using DISTRUCT v.1.1 [63]. A genotype was considered to belong to a group if its membership coefficient was ≥ 0.70. Genotypes with membership coefficient lower than 0.70 at each assigned K were regarded as admixed.

To cross-check the results from the model-based population structure from STRUCTURE with a model-free other method, DAPC was used. DAPC is a multivariate method designed to identify and describe clusters of genetically related individuals [64]. In the absence of a known grouping pattern, DAPC uses sequential K-means and model selection to build genetic clusters based on information from genetic data. The Bayesian information criterion (BIC) was used to identify an optimal number of genetic clusters (K) to describe the data. Based on the calculation of the α-score, the optimal number of principal components was retained. DAPC also provides membership probabilities for each individual to each identified group (or subpopulation), which can be equated to admixture proportions provided by STRUCTURE [59].

PCoA is a distance-based approach to dissect and display dissimilarities between individulas. The number of clusters obtained from STRUCTURE and DAPC was compared with those from PCoA without any assumption about the underlying population genetic model and it was performed using the dartR-R package [65].

## Construction of core collection

DARwin version 6.0.010 was used to build the diversity trees [66]. Dissimilarities were calculated and transformed into Euclidean distances. Un-Weighted Neighbor-Joining (N-J) method was applied to the Euclidean distances to build a tree with all genotypes. Then, 'maximum

length sub tree function' was used to draw the core collection. Maximum length sub-tree implemented is a stepwise procedure that successively prunes redundant individuals. This procedure allows the choice of the sample size which retains the largest diversity and is visualized by the tree as built on the initial set of accessions (357 genotypes). The size of the core collection and efficiency of the strategy was assessed by comparing and keeping the total number of alleles captured for each run using the same software. The size of the core collection was expressed as a proportion of the number of individuals selected for the core collection to the number of individuals in the entire collection.

## Results

### SNP variations

Of the total of 20 276 SNP markers, 6 498 (32%) SNP markers and 357 cowpea genotypes were retained after filtering. These 6 498 SNP markers were spread over the 11 chromosomes with an average of 591 markers per chromosome (S2 Table). Among the 11 chromosomes, the overall polymorphic information content values ranged from 0.25 (chromosome 9) to 0.30 (chromosome 6), with an average of 0.28. Gene diversity (Ho) values varied from 0.30 on chromosome 9 to 0.36 on chromosome 6, with an average of 0.33. The expected heterozygosity (He) values ranged from 0.11 (chromosome 9) to 0.13 (chromosome 4, 6 and 11), with an average of 0.12. For all chromosomes, the expected heterozygosity values (He) were higher than the observed heterozygosity values (Ho).

In the collected cowpea genomes, more transition-type SNPs (57%) were observed than transversion-type SNPs (43%), with a transition/transversion (Ts/Tv) SNP ratio of 1.33:1 (3714/2784). More A/G and C/T transitions were observed than G/A and T/C transitions. On the other hand, more G/T, A/T, A/C and C/G transversions were observed than T/G, T/A, C/A and G/C transversions.

### Genetic diversity and relationship

The genetic parameter estimate of the pre-defined subpopulations is presented in Table 1. Landraces collected from Tigray, Amhara and Oromia had a higher Shannon diversity index (H' = 0.44–0.45) and higher gene diversity (He = 0.29–0.30) than those from Gambella and SNNPRS. The population from Tigray had a higher heterozygosity value (Ho = 0.21) than those from other regions, whereas those from Gambella and SNNPRS had the lowest observed and expected heterozygosity values (Ho = 0.11). Landraces collected from the Oromia region had the highest inbreeding coefficient (FIS = 0.55) and the population from Tigray a relatively low inbreeding coefficient (FIS = 0.28), suggesting that 62% of the alleles were not fixed in the latter. Improved cultivars showed lower diversity (0.24) compared to the whole landraces population (0.30), with lower inbreeding in landraces (0.55) than in the improved varieties (0.58). The same trend was observed whie comparing the improved cultivars to the subpopulations of Oromia and Tigray that have similar size.

Genetic distance among cowpea genotypes varied from the lowest of 0.00 to the highest of 0.69 based on Euclidean distances. Twenty-nine pairs of genotypes had a genetic distance of 0, suggesting that the members of these pairs may in fact have been separately collected, but had an identical genetic background. Six pairs of genotypes, with an average genetic distance of 0.69, were found to be highly divergent. Cowpea landrace CP20, which was collected from Tigray, was found to diverge highly from other landraces, including CP5 (SNNPRS), CP259 (Amhara), CP277 (Oromia) and CP352 (SNNPRS). Another landrace, CP23, from the Amhara region was found to diverge from CP5 (SNNPRS) and CP259 (Amhara). The resulting distance matrix was used to construct an UPGMA dissimilarity dendrogram that classified the 357

**Table 1. Genetic parameter estimates based on 6498 SNPs among cowpea subpopulations.**

| Populations | Genetic Parameters | | | | |
|---|---|---|---|---|---|
| | N | H' | $H_o$ | $H_e$ | $F_{IS}$ |
| **Based on Geographical Region** | | | | | |
| Amhara | 70 | 0.44 | 0.13 | 0.29 | 0.51 |
| Gambella | 59 | 0.41 | 0.12 | 0.27 | 0.47 |
| Oromia | 49 | 0.44 | 0.12 | 0.29 | 0.55 |
| SNNPRS | 92 | 0.41 | 0.11 | 0.27 | 0.52 |
| Tigray | 41 | 0.45 | 0.21 | 0.30 | 0.28 |
| Mean | | 0.43 | 0.14 | 0.28 | 0.47 |
| SE | | 0.008 | 0.018 | 0.006 | 0.048 |
| **Based on Breeding Status** | | | | | |
| Improved | 47 | 0.37 | 0.06 | 0.24 | 0.58 |
| Landraces | 311 | 0.47 | 0.13 | 0.30 | 0.55 |
| Mean | | 0.42 | 0.10 | 0.27 | 0.57 |
| SE | | 0.05 | 0.05 | 0.03 | 0.02 |
| Overall mean | | 0.43 | 0.12 | 0.28 | 0.52 |
| SE | | 0.004 | 0.02 | 0.004 | 0.04 |

N, Number of genotypes; H', Shannon diversity index; Ho, heterozygosity; He, gene diversity; FIS, inbreeding coefficient.

cowpea genotypes into three distinct clusters (Fig 2). The first cluster (C-1) contained 31% of the genotypes (88 landraces and 22 improved varieties), the second (C-2) contained 24% of the genotypes, all of them being landraces with a major contribution from Amhara and Tigray, and the third cluster (C-3) contained 45% of the genotypes, including 25 improved varieties and 136 landraces mostly from Gambella, SNNPRS and Oromia. The improved varieties showed a clear tendency to be clustered together into only two clusters (C-1 and C-3).

The neighbor joining (NJ) tree constructed from the pairwise FST values also grouped the regions of origin into three distinct clusters (Fig 3). The adjacent regions showed tendencies to be grouped together. Gambella and SNNPRS are in the first group (P-I); Oromia, Amhara and Tigray in the second group (P-II) and the introductions are in the third group (P-III) (Fig 3).

## Analysis of molecular variance

The analysis of molecular variance of the 357 cowpea genotypes as pre-defined, based on geographical regions and breeding status, is presented in Table 2. The results indicated high variance within populations of 46% based on geographical regions and 53% based on breeding status. Likewise, variation within individuals of 44% was recorded based on geographical regions and 39% based on breeding status. Variance among populations based on geographical regions accounted for only 10% of the variation and 8% based on breeding status.

A high inbreeding coefficient ($F_{IS}$), overall fixation index ($F_{IT}$) and genetic differentiation ($F_{ST}$) values were observed. The $F_{IT}$ values for all the SNP loci ranged between 0.641 and 0.720, and $F_{IS}$ values between 0.602 and 0.698 based on geographical regions and breeding status, respectively. The pairwise FST values for all the SNP loci showed significant differentiation, ranging from -0.011 to 0.410 and -0.012 to 0.718 among the subpopulations, based on geographic regions and breeding status, respectively. The analysis showed low to moderate differentiation in allele frequencies among the populations ($F_{ST}$), 0.10 and 0.07, based on geographical area and breeding status, respectively. Gene flow between and within geographical regions was also moderate (Nm = 2.409).

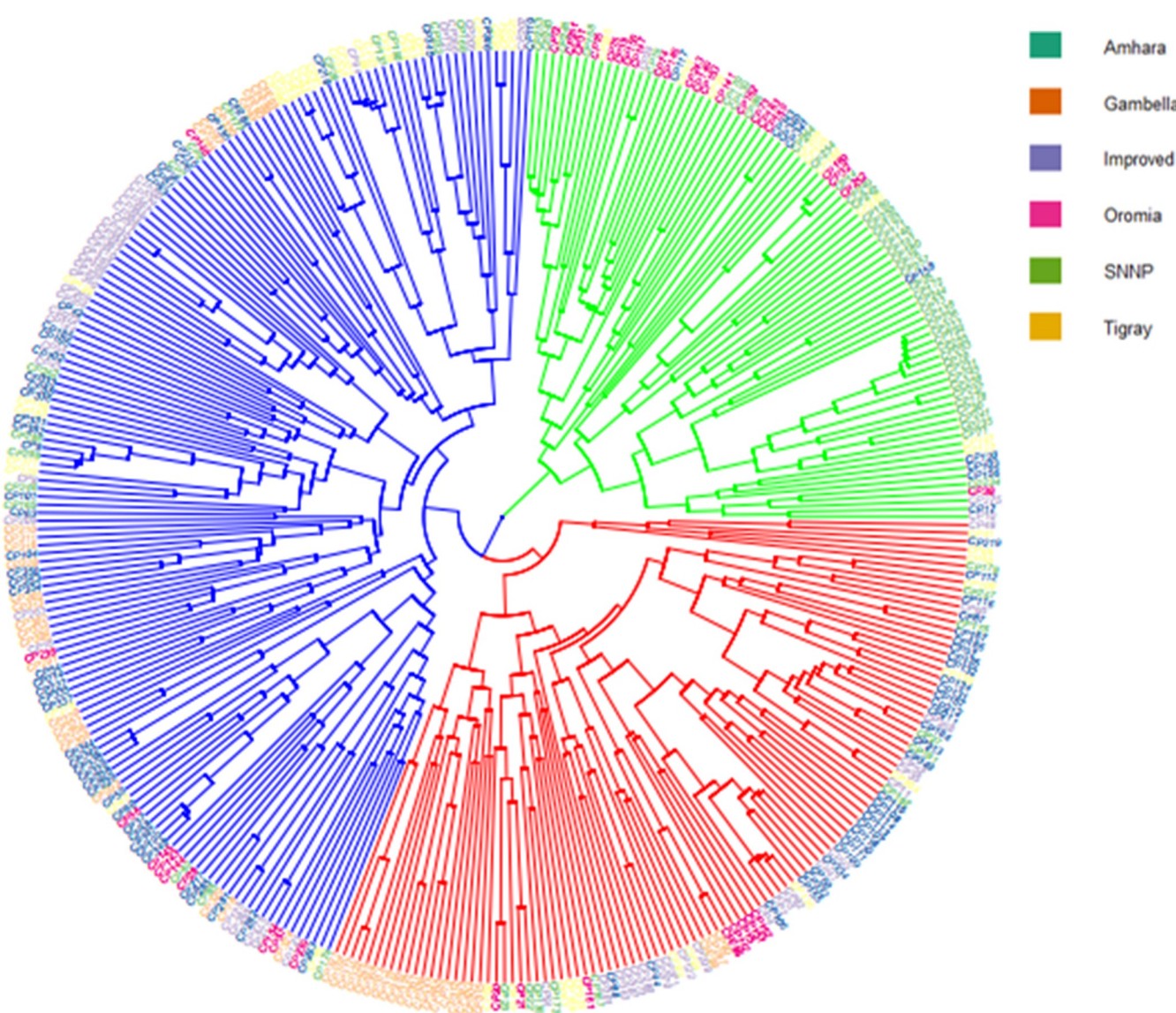

**Fig 2. UPGMA dendrogram showing the genetic relationships among cowpea collections grouped into three distinct clusters (cluster 1 = red, cluster 2 = green and cluster 3 = blue).**

## Population structure

The population structure of the 357 cowpea genotypes was determined using STRUCTURE software. The most probable number of subpopulations in the collected cowpea genotypes was K = 3. Based on the probable likelihood of each genotype to be grouped into any of the three distinct groups, 57 (16%) fell into the first cluster (C-I), 55 (13%) into the second cluster (C-II), and 124 (34%) into the third cluster (C-III) (Fig 4). The remaining 121 of the 357 accessions (36%) were placed in the admixture group (Table 3). All five subpopulations of landraces based on the geographical regions had three structured populations, whereas the improved cultivars had shown only two (C-I and C-III).

To confirm the true value of K, another model-free method, DAPC, was used. The optimum number of clusters was obtained with K = 3 using the BIC, which again divided the

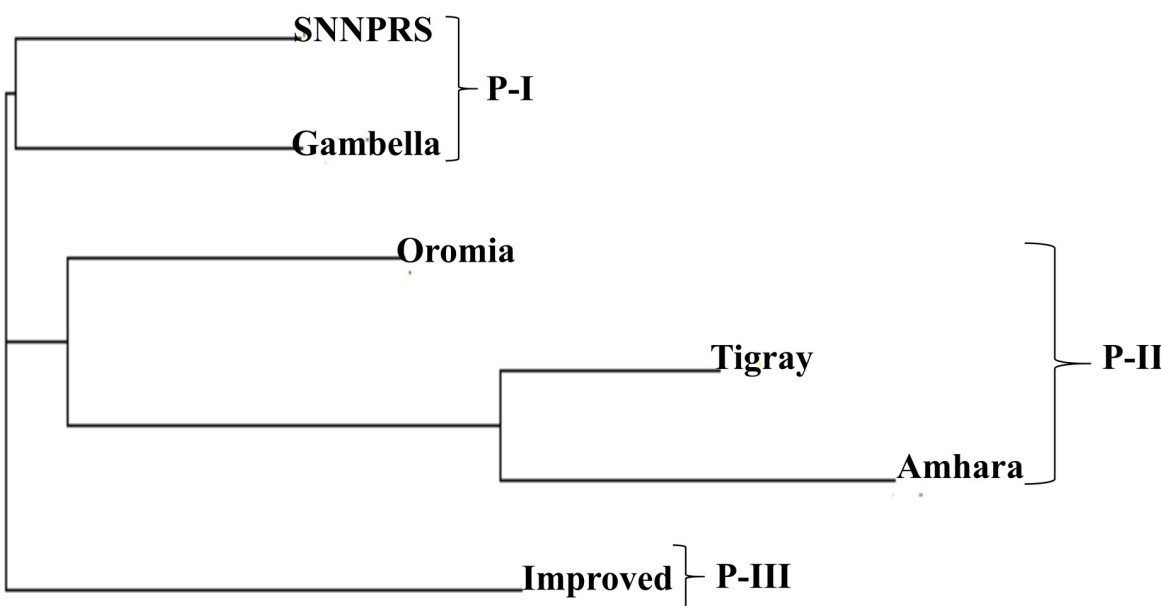

**Fig 3. Neighbor joining tree among five different regions of Ethiopia and introduced improved cultivars based on pairwise FST.**

genotypes into three sub-populations. Membership clustering using DAPC also grouped the genotypes into three clusters (Fig 5). The first cluster had 109 (31%) genotypes, of which 22 were improved varieties and 87 landraces. The second cluster had 86 genotypes, which were all landraces and the third cluster had 162 genotypes, including 25 improved varieties and 137 landraces (Table 3).

PCoA showed that the first three principal component vectors explained a total of 37.8% of the genotypic variability, of which 32.9% was contributed by the first two principal components (PC1 and PC2). A bi-plot of the first two principal components (PC1 and PC2) also revealed a more or less consistent population structure, as presented earlier (Fig 6).

Core collection. A total core set of 94 individuals out of 357 genotypes were sufficient to retain 100% of SNP diversity and captured all the alleles revealed by the 6498 SNPs. The core collection comprised 71 landraces collected from different regions of Ethiopia and 23

**Table 2. Analysis of molecular variance among and within cowpea subpopulations.**

| Source of variation | Df | SS | MS | EV | PV | F-Statistic |
|---|---|---|---|---|---|---|
| **Based on Geographical Origin** | | | | | | |
| Among populations | 4 | 90397 | 22599 | 167 | 10 | $F_{ST} = 0.01$ |
| Within populations | 306 | 700158 | 2288 | 775 | 46 | $F_{IS} = 0.01$ |
| Within individuals | 311 | 229322 | 737 | 737 | 44 | $F_{IT} = 0.01$ |
| Total variations | 621 | 1019877 | 1642 | 1679 | 100 | |
| **Based on Breeding Status** | | | | | | |
| Among populations | 1 | 24591 | 24591 | 135 | 8 | $F_{ST} = 0.01$ |
| Within populations | 356 | 883411 | 2481 | 904 | 53 | $F_{IS} = 0.01$ |
| Within individuals | 358 | 241095 | 673 | 673 | 39 | $F_{IT} = 0.01$ |
| Total variations | 715 | 1149097 | 1607 | 1713 | 100 | |

Df = degrees of freedom; SS = sum of squares; EV = estimated variance, PV = percentage variance; $F_{ST}$ = genetic differentiation, $F_{IS}$ = fixation index or inbreeding coefficient and $F_{IT}$ = Overall fixation index.

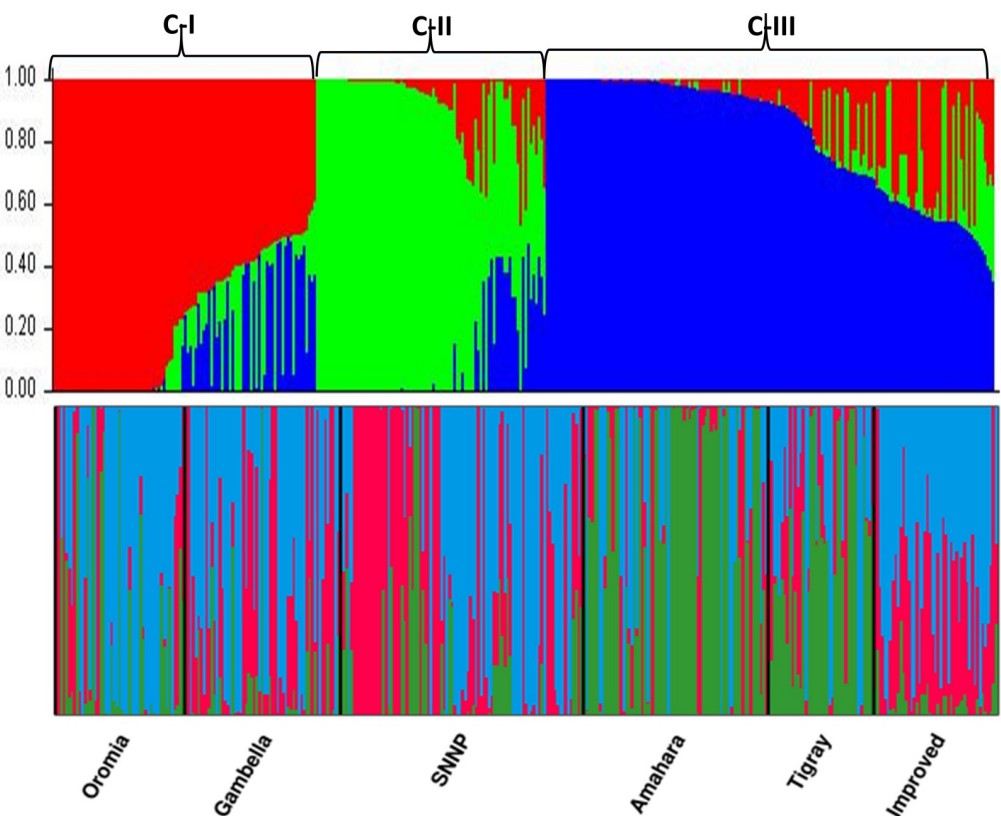

**Fig 4. Population structure of 357 cowpea genotypes, in K = 3; each color represents one cluster.**

**Table 3. Proportion of membership of each predefined population in each of the clusters obtained at the best K (K = 3).**

| Population | Number of accessions | Admixed individual | Proportion of membership in each cluster (%) | | |
|---|---|---|---|---|---|
| | | | Cluster I | Cluster II | Cluster III |
| **STRUCTURE** | | | | | |
| Amhara | 70 | 23 | 6 | 53 | 19 |
| Gambella | 59 | 29 | 22 | 3 | 46 |
| Improved | 47 | 53 | 9 | 0 | 38 |
| Oromia | 49 | 27 | 8 | 12 | 53 |
| SNNPRS | 92 | 29 | 30 | 2 | 38 |
| Tigray | 40 | 58 | 5 | 25 | 13 |
| | **357** | **36.4** | **13.3** | **15.9** | **34.4** |
| **DAPC** | | | | | |
| Amhara | 70 | 15 | 10 | 59 | 16 |
| Gambella | 59 | 32 | 25 | 2 | 41 |
| Improved | 47 | 13 | 40 | 0 | 47 |
| Oromia | 49 | 31 | 10 | 10 | 49 |
| SNNPRS | 92 | 17 | 41 | 3 | 39 |
| Tigray | 40 | 30 | 12 | 38 | 20 |
| | **357** | **23.0** | **23.0** | | **35.3** |

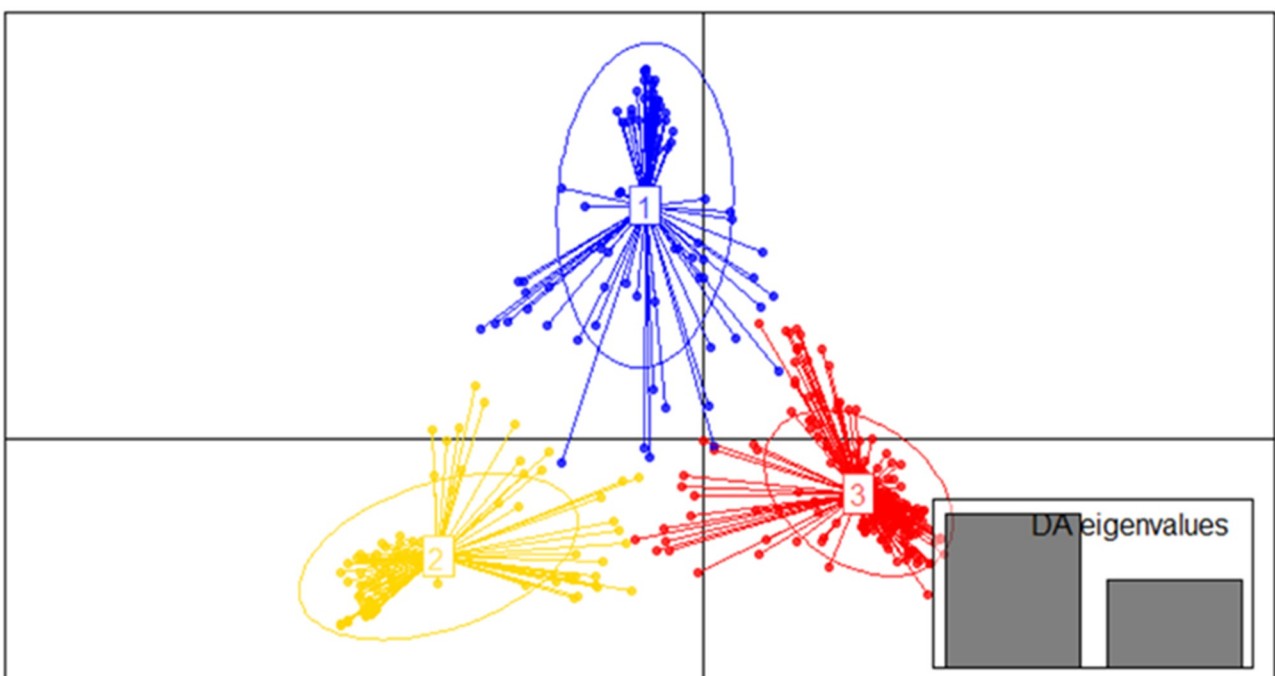

**Fig 5. Scattered plot of DAPC, each color represents one cluster.**

genotypes drawn from the improved cowpea cultivars representing 76% and 24% of the core collection respectively (Table 4). The core collection named as CC-94 represents 26% of the entire collections. In the core collection all the geographical regions were represented by 20% to 33% of genotypes from the total genotypes collected (Table 4). Fourteen [14], 15, 23, 11, 18

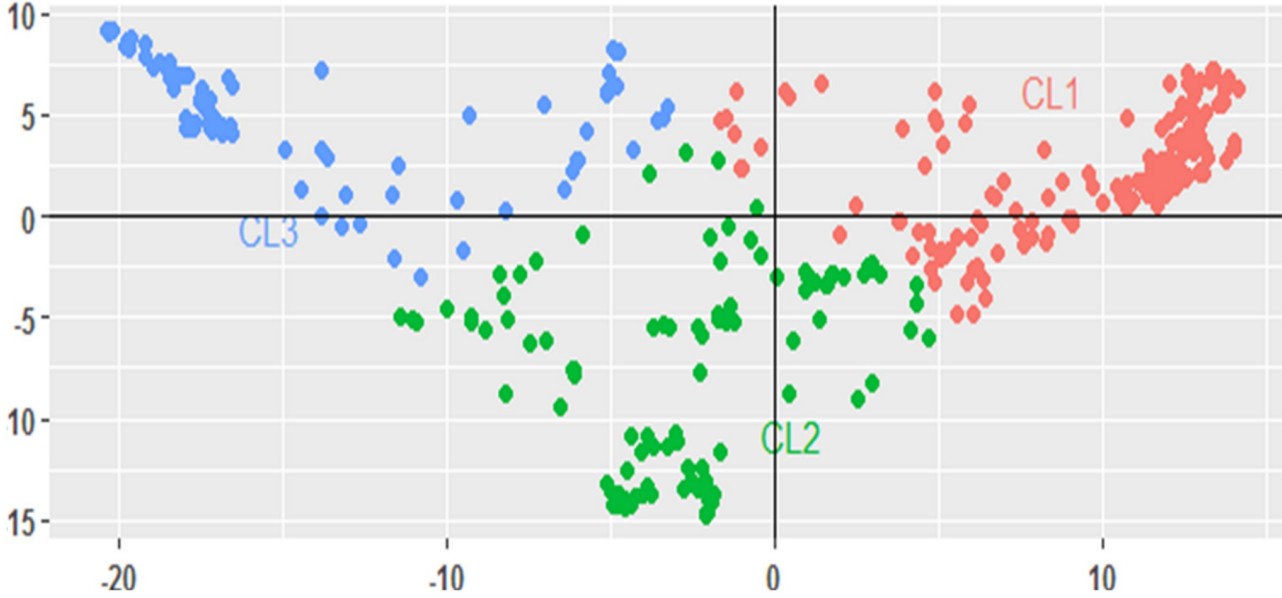

**Fig 6. A bi-plot of the first two principal components (PC1 and PC2) of 357 cowpea genotypes, using 6 498 SNP markers.** Each color corresponds to population structuring and grouping.

**Table 4. Number and proportion of accessions in the core collection along with each collection regions.**

| Collection regions | Number of accessions by region | Number of accessions from the core collection by region | % of the core collection along region |
|---|---|---|---|
| Amhara | 70 | 14 | 20 |
| Gambella | 59 | 15 | 25 |
| Improved | 47 | 23 | 49 |
| Oromia | 49 | 11 | 22 |
| SNNPRS | 92 | 18 | 20 |
| Tigray | 40 | 13 | 33 |
| Total | 357 | 94 | 26 |

and 13 accessions were drawn from Ambara, Gambella, Improved cultivars, Oromia, ANNPRS and Tigray respectively. These accessions represent 20%, 25%, 49%, 22%, 20% and 33% respectively of the total individuals investigated from each region (Table 4).

## Discussion

### SNP variation

The genetic diversity in a set of 357 Ethiopian cowpea genotypes as revealed by 6 498 SNP markers showed all possible types of SNPs in these genotypes, A/G and C/T, being most prevalent. Observation of a transition/transversion (Ts/Tv) ratio of 1.33:1 may reflect high frequencies of A to G and C to T mutations following methylation. This result is in agreement with what was found in a global collection of 422 cowpea landraces and African ancestral wild cowpea genotypes previously genotyped with 1 536 SNPs markers [67] and another study of 768 cultivated cowpea genotypes from the USDA GRIN cowpea collections, genotyped with 1048 SNP markers [47].

### Genetic diversity and relationship

The landraces collected from different parts of Ethiopia showed 13% allele heterozygosity compared to only 6% in improved varieties, indicating that most of the alleles in improved cowpea varieties were almost fixed to homozygosity. The Shannon-Weaver index is highly correlated, with evenness, i.e. the number of genotypes per population and/or number of unique genotypes represented in the population [68,69]. Despite the relatively small number, landraces collected from the Tigray region [41] showed the highest values of gene diversity (0.30) and heterozygosity (0.21). On the contrary, landraces collected from SNNPRS had the largest relative sample size [92] and showed the lowest gene diversity (0.27) and heterozygosity (0.11). Our result is different from that reflected in a previous report based on 81 Ethiopian cowpea accessions analyzed using SSR markers, which stated that accessions from the SNNPRS region had higher genetic diversity than those from Tigray [30]. This discrepancy might be due to a very small number of samples from the Tigray region [6] in the previous report, or the difference in the type of markers used, or both. It is believed that the differentiation of genotypes into different clusters is independent of the type of germplasm [70], the type of marker used [71,72], the primers selected within markers [72,73] and the sampling strategy [73].

The gene diversity values obtained for the entire population are similar to those reported from the world's largest cowpea collections using SNP markers [47,48]. A study from Africa of Senegalese cowpea varieties using SSR markers showed similar results as those found in the present study [16], but higher values were observed compared to Chinese cowpea collections genotyped using SSR markers [74] and Iberian Peninsula collections with SNP markers [75].

Results from the present study, however, showed the existence of lower genetic diversity compared to similar studies in many other self-pollinated crops such as common bean [76], and chickpea and lentil [77]. This could be corroborated by the fact that, in general, cowpea has a narrow genetic base due to the initial bottleneck during domestication [3,4,36,41], and strict self-pollinated nature of the crop [78–80].

It is worth noting that the cowpea landraces used in this study were collected from farmers' fields and local markets; most of the cowpea growers in Ethiopia use farm-saved seeds of their own [81] and this type of seed system limits the movement of germplasm from farmer to farmer and among localities. Cowpea breeding is also still in its early infancy in Ethiopia and all these considerations together limit the development and provision of varietal options to the farmers, which, in turn, may inhibit the integration of new genotypes from other sources and result in limited genetic diversity of the crop.

## Analysis of molecular variance

Variations within populations and within individuals accounted for the largest proportion of the total variation. Coupled with the lower fixation index ($F_{ST}$) estimates (0.10 and 0.07) and the small percentage of variation among populations (10% and 8% based on collection region and breeding status, respectively), this may suggest a low to moderate level of differentiation among populations with an increased level of admixtures. In practice, an $F_{ST}$ of 0.00–0.05 indicates low differentiation, 0.05–0.15 indicates moderate differentiation and 0.15–0.25 high levels of differentiation, while an $F_{ST} > 0.25$ indicates a very high level of differentiation [82–84]. As stated earlier, this lower level of variation among populations might be attributed to germplasm exchange among regions, limited introduction of new varieties to the farming system in each region and wider agro-ecological adaptation of the crop. This result is in agreement with many diversity studies in cowpea collections using different markers [30,48,67,74,85,86].

In the Ethiopian cowpea collection, we found pairwise $F_{ST}$ values ranging from 0.022 to 0.122, indicating low to moderate levels of genetic differentiation among regions. Wright indicated that if Nm > 1 [87], there is enough gene flow. The gene flow ($N_m$) among regions in the current study is 2.409, indicating the existence of germplasm exchange among cowpea accessions collected from different geographic regions and the introduced improved cowpea cultivars. Similar results were reported from a previous study on Ethiopian cowpea germplasm collections [30].

## Population structure

Information about the structure of germplasm collections is of great importance for both conservation and utilization of genetic resources. Different approaches were used to infer the population structure of Ethiopian cowpea germplasm; the Bayesian model-based clustering algorithm using STRUCTURE and DAPC and the optimal K value were compared with those from principal coordinate analysis and the UPGMA tree. These methods showed the existence of three main ancestral populations. DAPC is a clustering multivariate method that uses sequential K-means and model selection [57] for genetic clustering in the absence of a prior grouping pattern. It provides an interesting alternative to STRUCTURE software, as it does not require that populations are in Hardy-Weiber equilibrium and can handle large sets of data without using parallel processing software. Nevertheless, our results showed good consistency between STRUCTURE and DAPC analyses when no admixed individuals were considered. The result is in agreement with a previous structuration of Ethiopian cowpea genotypes using the SSR marker [30], as indicated by previous works from the world collection [47,48].

Despite the availability of newer approaches, traditional hierarchical clustering analysis such as UPGMA provides easy and effective determination of genetic diversity in plants [88]. Furthermore, multivariate relationships among accessions were revealed through PCoA. Both the UPGMA tree and PCoA confirm the result from STRUCTURE and DAPC.

The clustering of the genotypes presented in this study may give interesting clues for increasing diversity in breeding programs and germplasm collections. Landraces were spread in all three clusters, whereas most of the improved cultivars were included in only two clusters (C-2 and C-3). Hence, the use of landraces different from clusters 2 and 3, as founding clones, may increase the genetic diversity of new cultivars. Deep knowledge of the population structure and understanding of the clustering pattern would assist the efficient choice of parental lines in current breeding programs. This will maximize genetic diversity, enhance the potential gain from selection and would help to increase the breeding programs' efficiency to face new demands from producer, consumers, and emerging ecological constraints (i.e. adaptation to climate change, resistance against pests).

## Core collection

Establishement of core collection is important to have manageable and representative sample size that can represent the diversity of the entire collection [89]. Brown believes that a core collection sample size from 5% to 10% of the original germplasm resources can represent more than 70% of the genetic variations of the whole germplasm [31]. However, Yonezawa et al. [90] assumed 20%–30% of the sampling percentage was needed to well conserve the genetic diversity of the entire collection. This trend was observed in our study, where 26% of genotypes were sufficient to retain 100% of SNP diversity of the whole population. This also indicated the existence of remarkable genetic diversity in the Ethiopian cowpea collection. Similar result (27%) was reported in faba bean [91]. Conversely, lower proportion of core collection was observed in lupin [92] at 16% while higher proportion of 36% as core collection size was reported in common bean [93]. The present result demonstrated the potential of highly informative and selective DArTSeq SNP markers to construct core collection and to enhance proper utilization and conservation of Ethiopian cowpea accesstions. This core collection will serve as a primary source for SNP mining and further associational analysis for novel genes in cowpea.

## Conclusion

Three well-differentiated genetic populations or clusters were postulated from this study in the 310 Ethiopian cowpea landraces and 47 improved cultivars based on genome-wide SNPs scanning. This population structure will inform a genomic selection-based approach to introgress genomic regions associated to ion content both in the leaf and grain of cowpea. Although different reports indicated that East Africa, including Ethiopia, is one of the centers of origin, center of diversity or secondary center of diversity, there was no cowpea collection neither any characterization to such extent involving Ethiopia cowpea landraces. Though this collection is only limited to Ethiopia, the result of this study shed light on the existence of genetic diversity in the landraces more than the cowpea collection used worldwide and it's expected that these landraces might have unharnessed potential for future breeding owing different traits for cowpea improvement. Thus, these germplasms can be used globally for cowpea future breeding.

The limited genetic distances among the pre-defined populations suggest the existence of a large number of duplications of accessions in the Ethiopian cowpea germplasm collections. Therefore, a core collection was built to avoid genotype duplication in germplasm. In this study, we propose the first core collection of 94 accessions capturing all the diversity from the

Ethiopian landrace collections representing five administrative regions and 47 improved culti-vars from international research centers, based on GBS derived SNP markers. The establish-ment of this core collection will also enhance the proper conservation and utilization of cowpea genetic resources for crop improvment. The present study also demonstrated the potential of highly informative and selective DArTSeq-derived SNP markers for genetic diver-sity and population structure studies. Therefore, given the proven high variability level of the Ethiopia germplam, a core set of diagnostic marker for cowpea genetic diversity study can be built for use worldwide and in Ethiopia. Similarly, Kompetitive Allele Specific PCR markers, derived from the GBS (DArTSeq)-SNPs, can be designed, validated and used for marker trait association of the crop such as calcium, iron, magnesium and zinc content for traits that are important for the economy and nutrition.

## Supporting information

**S1 Table. Collection region, number of accession from each region and altitudinal ranges of the test cowpea genetic resources.**
(DOCX)

**S2 Table. Distribution and genetic diversity parameters of 6498 SNPs measured in a set of 357 cowpea genotypes.**
(DOCX)

**S1 File. GPS coordinates data used to construct the map showing the collection sites (Fig 1).**
(XLSX)

**S2 File. Genotyping supporting data after filtering used for the study.**
(TXT)

## Acknowledgments

The study was part of PhD research work of the first author (SK) and she is grateful to the Ethi-opian Institute of Agricultural Research for giving her study leave. The authors are grateful to the Collaborative Crop Research Project of McKnight Foundation for the collection and main-tenance of the accession and germplasm respectively; and the Ethiopian Bio-diversity Institute for provision of some of the accessions.

## Author Contributions

**Conceptualization:** Selamawit Ketema, Bizuayehu Tesfaye, Gemechu Keneni, Berhanu Amsalu Fenta, Nasser Yao.

**Data curation:** Selamawit Ketema.

**Formal analysis:** Selamawit Ketema, Ermias Assefa, Nicolas Greliche, Eunice Machuka, Nas-ser Yao.

**Funding acquisition:** Selamawit Ketema, Nasser Yao.

**Investigation:** Selamawit Ketema.

**Methodology:** Selamawit Ketema, Bizuayehu Tesfaye, Gemechu Keneni, Berhanu Amsalu Fenta, Ermias Assefa, Nicolas Greliche, Eunice Machuka, Nasser Yao.

**Project administration:** Selamawit Ketema, Nasser Yao.

**Resources:** Selamawit Ketema, Bizuayehu Tesfaye, Gemechu Keneni, Berhanu Amsalu Fenta, Ermias Assefa, Nicolas Greliche, Nasser Yao.

**Supervision:** Bizuayehu Tesfaye, Gemechu Keneni, Berhanu Amsalu Fenta, Nasser Yao.

**Visualization:** Selamawit Ketema.

**Writing – original draft:** Selamawit Ketema.

**Writing – review & editing:** Bizuayehu Tesfaye, Gemechu Keneni, Berhanu Amsalu Fenta, Ermias Assefa, Nicolas Greliche, Eunice Machuka, Nasser Yao.

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
