## [Decision Letter · Decision Letter 0]

7 Apr 2020

PONE-D-19-32934

DArTSeq SNP-based markers revealed high genetic diversity and structured population in Ethiopian cowpea [Vigna unguiculata (L.) Walp] germplasms

PLOS ONE

Dear Dr. Yao,

Thank you for submitting your manuscript to PLOS ONE. After careful consideration, we feel that it has merit but does not fully meet PLOS ONE’s publication criteria as it currently stands. Therefore, we invite you to submit a revised version of the manuscript that addresses the points raised during the review process.

We would appreciate receiving your revised manuscript by May 22 2020 11:59PM. To enhance the reproducibility of your results, we recommend that if applicable you deposit your laboratory protocols in protocols.io, where a protocol can be assigned its own identifier (DOI) such that it can be cited independently in the future. For instructions see: http://journals.plos.org/plosone/s/submission-guidelines#loc-laboratory-protocols

We look forward to receiving your revised manuscript.

Kind regards,

Tzen-Yuh Chiang

Academic Editor

PLOS ONE

Journal Requirements:

https://bmcplantbiol.biomedcentral.com/articles/10.1186/s12870-016-0712-9

In your revision ensure you cite all your sources (including your own works), and quote or rephrase any duplicated text outside the methods section. Further consideration is dependent on these concerns being addressed.

Reviewers' comments:

Reviewer's Responses to Questions

**Comments to the Author**

1. Is the manuscript technically sound, and do the data support the conclusions?

Reviewer #1: Partly

Reviewer #2: Yes

2. Has the statistical analysis been performed appropriately and rigorously? 

Reviewer #1: Yes

Reviewer #2: Yes

3. Have the authors made all data underlying the findings in their manuscript fully available?

Reviewer #1: No

Reviewer #2: Yes

4. Is the manuscript presented in an intelligible fashion and written in standard English?

Reviewer #1: Yes

Reviewer #2: No

5. Review Comments to the Author

Reviewer #1: Dear Editor and Author:

In the article “DArTSeq SNP-based markers revealed high genetic diversity and structured

population in Ethiopian cowpea [Vigna unguiculata (L.) Walp] germplasms”, the authors evaluated the genetic diversity in 357 Ethiopian cowpea germplasm, using 6498 GBS based single nucleotide polymorphism (SNPs) markers. The population structure, genetic diversity, and phylogeny were analyzed in this research.

Generally, this manuscript has a large amount of data, a clear idea, and a relatively intelligent data analysis. We can see the author's research ability and efforts. Personally, I like this article very much. However, I think this manuscript is not appropriate to be published right now, until the below problems are solved:

1. This manuscript is to describe the diversity of cowpea in Ethiopian, why the author added breeding line and varieties which is not collected from Ethiopian? I agree and recommend adding a certain amount of breeding lines for diversity analysis, however, the source of the materials does not associate with the main idea of the article, it will cause confusion in the analysis. If the author implied or thought those breeding lines were all from Ethiopia, why compare breeding lines with all Ethiopian landraces as two independent groups in Table 1 and 2?

2. How did the authors pick up the 314 landraces from whole germplasms? Are these materials representative of the entire germplasm pool? Theoretically, diversity analysis must collect all local landraces, but considering the research cost. The researchers will select some highly diverse materials from all the collections to represent the whole germplasm pool. Please clarify whether these materials have such qualifications. My suggestion is that if the 314 landraces are part of germplasm, the author should provide the phenotype of all germplasm resources and prove the representativeness of the selected material; if this material is the entire germplasm of Ethiopia, it is also desirable to conduct analysis and statistics on the phenotype of this material.

3. How did the authors screen the 6498 markers from the 20276 SNPs? The main problem of GBS is missing and false data, please clarify the quality control of genotype. In my suggestion, it is also better to prove the distribution of the 6498 SNPs in each chromosome, because of the character of GBS sequencing, the average distance of markers is too much different from the real situation.

4. Please re-make Fig 2, 5 and 6 to improve the resolution, if the authors could.

Except for the flaw of the scientific part, my main concerns are:

1 The germplasm in this manuscript was only from one country, which makes it of limited reference value to researchers worldwide. Please elaborate on the implications of this study for researchers worldwide.

2 Overall, although the diversity analysis of this study was relatively integrated, I did not see any innovation, depth understanding, or thinking about diversity research from the conclusions and perspectives.

To sum up, I am conservative and cautious about the publication of this manuscript immediately, unless the authors can find new research ideas or impressive conclusion with a more thorough discussion and understanding from current results. My suggestion is that, since the author has mentioned the establishment of a core collection, it is better to build the “core germplasm” for Ethiopia based on the existing data combining with the phenotype of whole the germplasm. This project is more important than an only diversity study, and its results are more valuable for reference over the world.

Sincerely

Reviewer #2: In this manuscript the authors explored the genetic diversity and population structure of Ethiopian cowpea using DArTSeq markers. However, the paper needs improvement. The authors forgot to include line numbers in the manuscript which make it difficult to mention typographical errors. A correction of the text is recommended.

In the abstract replace 357 cowpea “germplasms” by 357 cowpea accessions

In the introduction the 1st sentence needs correction, also there is a lack of consensus on where in Africa cowpea was domesticated. Authors should take that in account.

Methods section: a brief description of the sequencing method is recommended.

Authors should provide detailed information on the SNP calling approach, SNP filtering thresholds as well as the filtering of the accessions.

Population structure analysis: I would recommend a membership coefficient of ≥ 0.70.

Authors should also replace the figures with high resolution figures.

6. PLOS authors have the option to publish the peer review history of their article (what does this mean?). If published, this will include your full peer review and any attached files.

Reviewer #1: No

Reviewer #2: No

---

## [Author Response · Author response to Decision Letter 0]

14 Aug 2020

Rebuttal letter in response to the comments forwarded by the reviewers on manuscript number, PONE-D-19-32934 [DArTSeq SNP-based markers revealed high genetic diversity and structured population in Ethiopian cowpea [Vigna unguiculata (L.) Walp] germplasms].

Dear Editor and Reviewers

We are grateful for the reviewers who have been reviewing our manuscript during this COVID -19 pandemic time. We are also thankful for the valuable comments given to improve our manuscript. We would also like to seek apology for the delayed response for this paper, this was due to communication gap among the authors worsened by the pandemic situation and the internet restriction by the Ethiopian gouvernement. The reviewer’s comments have helped us to improve the manuscript and we have tried our best to address all the comments given by the reviewers. The changes made in the manuscript are shown in track in the manuscript. The response for each comment, along with the query of the reviewer is summarize as below. 

Sincerely, 

The Authors

Revision notes

Reviewer 1 Comments:

Comment 1. This manuscript is to describe the diversity of cowpea in Ethiopian, why the author added breeding line and varieties which is not collected from Ethiopian? I agree and recommend adding a certain amount of breeding lines for diversity analysis, however, the source of the materials does not associate with the main idea of the article, it will cause confusion in the analysis. If the author implied or thought those breeding lines were all from Ethiopia, why compare breeding lines with all Ethiopian landraces as two independent groups in Table 1 and 2?

Response 1.). One of our objectives is to see if the landraces collected from different regions of Ethiopia maintained and used by the local farmers are different from the cowpea genotypes collected from IITA or developed by breeding program in Ethiopia. The inclusion of the breeding lines is driven by the fact that the cowpea-breeding program, being dependent on the introduced cultivars imported and/or from the International Institute of Tropical Agriculture (IITA), would represent exotic/external germplasm outside Ethiopia. Our goal in comparing breeding lines/improved variety from IITA with Ethiopian landraces is to see the allelic richness of these two groups and hence develop a better germplasm management strategy and for use for breeding in Ethiopia and worldwide. In addition, to mitigate the bias that can be introduced by the difference in number of samples, we have compared the improved varieties with each of the subpopulation that constitute the whole population for landraces.

Comment 2. How did the authors pick up the 314 landraces from whole germplasms? Are these materials representative of the entire germplasm pool? Theoretically, diversity analysis must collect all local landraces, but considering the research cost. The researchers will select some highly diverse materials from all the collections to represent the whole germplasm pool. Please clarify whether these materials have such qualifications. My suggestion is that if the 314 landraces are part of germplasm, the author should provide the phenotype of all germplasm resources and prove the representativeness of the selected material; if this material is the entire germplasm of Ethiopia, it is also desirable to conduct analysis and statistics on the phenotype of this material.

Response 2. The 314 landraces represent the entire collection of cowpea germplasm in Ethiopia. From this study we planned to only phenotype the core collection extensively for several key and economically important traits to cut on cost usually associated with efficient broad genetic stock/germplasm characterization. 

Comment 3. How did the authors screen the 6498 markers from the 20276 SNPs? The main problem of GBS is missing and false data, please clarify the quality control of genotype. In my suggestion, it is also better to prove the distribution of the 6498 SNPs in each chromosome, because of the character of GBS sequencing, the average distance of markers is too much different from the real situation. 

Response 3. The comment is accepted, and one paragraph was inserted in material and methods under the section “SNPs calling and data filtering” regarding data filtering to respond to the request query. Data filtering conditions were included as follow: SNP markers with high proportion of missing data (>10%), individuals who have high rates of genotype missingness (>10%) and rare SNPs with <5% minor allele frequency (MAF) were discarded from further analysis using R software (version 2.8.3). The most informative SNPs were selected based on a threshold PIC value equal or higher than 0.2. Finally 357 cowpea genotypes and 6498 (32%) of SNP markers were maintained for further analysis. 

In addition, the distribution of SNPs per chromosomes was provided in S2 Table. The number of SNPs ranges from 463 on chromosome 2 to 849 on chromosome 3 with a mean of 591 SNPs per chromosome

Comment 4. Please re-make Fig 2, 5 and 6 to improve the resolution, if the authors could 

Response 4. The comment is accepted, and new captures were done to improve the figure quality subsequently. The figures were also validated through the PLOS one PACE requirements. 

Comment 5. Except for the flaw of the scientific part, my main concerns are:

The germplasm in this manuscript was only from one country, which makes it of limited reference value to researchers worldwide. Please elaborate on the implications of this study for researchers worldwide. Overall, although the diversity analysis of this study was relatively integrated, I did not see any innovation, depth understanding, or thinking about diversity research from the conclusions and perspectives.

To sum up, I am conservative and cautious about the publication of this manuscript immediately, unless the authors can find new research ideas or impressive conclusion with a more thorough discussion and understanding from current results. My suggestion is that, since the author has mentioned the establishment of a core collection, it is better to build the “core germplasm” for Ethiopia based on the existing data combining with the phenotype of whole the germplasm. This project is more important than an only diversity study, and its results are more valuable for reference over the world.

Response 5. The queries and suggestion of comment 5 has been addressed as follows and included in the conclusion to show the importance of the current study and its potential use worldwide

Although different reports indicated that East Africa including Ethiopia is one of the centers of origin, center of diversity or secondary center of diversity there was no cowpea collection neither any characterization to such extent in Ethiopia cowpea landraces. Though this collection was only done in Ethiopia, the result of this study has shown the existence of genetic diversity in the landraces and it expected that these germplasms might have unharnessed potential for future breeding owing different trait for cowpea improvement. Thus, these germplasms can be used globally for cowpea future breeding. 

Innovation, depth understanding, or thinking about diversity research from the conclusions and perspectives.

Worldwide, germplasm collection has been done for most crops for a long period, and it is assumed as we already had enough collection. However, this study could show, as we have not still collected the germplasm resource of the world. Thus, this study sheds light on how we could still have several germplasm in most crops that needs collection, preservation, and use for future breeding. In this study in the future, the collected germplasm will be thoroughly phenotype, and used for cowpea improvement for the target traits. A core set of diagnostic makers can be developed to serve worldwide purpose

My suggestion is that, since the author has mentioned the establishment of a core collection, it is better to build the “core germplasm” for Ethiopia based on the existing data combining with the phenotype of whole the germplasm, 

• The suggestion for establishing core collection is well taken and included in the manuscript as shown in the introduction, material method, result, discussion and conclusion. However, we wouldn’t be able to phenotype the whole collection due to budget limitation, hence we will be focusing on phenotyping extensively the core collection for several traits 

Reviewer 2 Comments:

Comment 1. The authors forgot to include line numbers in the manuscript, which make it difficult to mention typographical errors. A correction of the text is recommended.

Response 1 Comment accepted and line numbers included 

Comment 2. In the abstract replace 357 cowpea “germplasms” by 357 cowpea accessions

Response 2. Comment accepted and correction made according to the suggestion 

Comment 3. In the introduction, the first sentence needs correction; also, there is a lack of consensus on where in Africa cowpea was domesticated. Authors should take that in account.

Response 3. Comment accepted and the sentence was rephrased as “Cowpea [Vigna unguiculata (L.) Walp., 2n = 2x = 22] originated and was domesticated in Africa though the exact location of origin of domestication is still a matter of speculation and different authors suggest different areas in Africa, Northeastern Africa including Ethiopia (1–4), Central Africa (5), South Africa (6), and West Africa (5,7,8). (Line 45 to line 48) 

Comment 4. Methods section: a brief description of the sequencing method is recommended.

Response 4. A brief description of the GBS protocol as developed and implemented by DArT was added to the Material and Method under the section “DNA extraction and sequencing”

Comment 5. Authors should provide detailed information on the SNP calling approach, SNP filtering thresholds as well as the filtering of the accessions.

Response 5. One paragraph was inserted in Material and Methods section with respect to data filtering under a new title “SNP calling and data filtering”

Comment 6. Population structure analysis: I would recommend a membership coefficient of ≥ 0.70.

Response 6. The comment accepted and membership coefficient of ≥ 0.70 were used and the results were corrected accordingly. 

Comment 7. Authors should also replace the figures with high resolution figures.

Response 7. The comment is accepted, and the figures have been replaced with figures with higher resolution quality. 

Response to reviewer comments, August 05

The reviewer comments are shown below in black and the response by the authors is provided as response to reviewers in italics

Dear Dr Yao,

Thank you for submitting your manuscript entitled "DArTSeq SNP-based markers revealed high genetic diversity and structured population in Ethiopian cowpea [Vigna unguiculata (L.) Walp] germplasms" to PLOS ONE. Your manuscript files have been checked in-house but before we can proceed we need you to address the following issues:

1) Thank you for updating your data availability statement. You note that your data are available within the Supporting Information files, but no such files have been included with your submission. At this time we ask that you please upload your minimal data set as a Supporting Information file, or to a public repository such as Figshare or Dryad.

 Please also ensure that when you upload your file you include separate captions for your supplementary files at the end of your manuscript.

As soon as you confirm the location of the data underlying your findings, we will be able to proceed with the review of your submission. 

Response 1 to reviewers: The genotyping data set was included as supporting information and a caption was added at the end of the manuscript under the title: S3: Genotyping supporting data obtained after filtering used for the study

2) Please amend your list of authors on the manuscript to ensure that each author is linked to an affiliation.

We note that you have included affiliation numbers 1-6 and *, ¶ however only affiliations 1-6 and * have authors linked to them. Please amend affiliation ¶ to link an author to it or remove if added in error.

Response 2 to reviewers: The affiliation “¶” was added by mistake and was removed from the author list

3) Please include a copy of Table 4 which you refer to in your text on page 10.

Response 3 to reviewers: Table 4 and Table 6 should read the same. The numbering as Table 6 came from the fact that S1 and S2 tables were previously included in the manuscript body as Table 4 and Table 5 before been moved as supplementary files. Then the actual Table 6 was amended as Table 4 in the table caption and in the text body.

---

## [Decision Letter · Decision Letter 1]

1 Sep 2020

DArTSeq SNP-based markers revealed high genetic diversity and structured population in Ethiopian cowpea [Vigna unguiculata (L.) Walp] germplasms

PONE-D-19-32934R1

Dear Dr. Yao,

We’re pleased to inform you that your manuscript has been judged scientifically suitable for publication and will be formally accepted for publication once it meets all outstanding technical requirements.

Kind regards,

Tzen-Yuh Chiang

Academic Editor

PLOS ONE

Additional Editor Comments (optional):

Reviewers' comments:

Reviewer's Responses to Questions

**Comments to the Author**

1. If the authors have adequately addressed your comments raised in a previous round of review and you feel that this manuscript is now acceptable for publication, you may indicate that here to bypass the “Comments to the Author” section, enter your conflict of interest statement in the “Confidential to Editor” section, and submit your "Accept" recommendation.

Reviewer #2: All comments have been addressed

2. Is the manuscript technically sound, and do the data support the conclusions?

Reviewer #2: Yes

3. Has the statistical analysis been performed appropriately and rigorously? 

Reviewer #2: Yes

4. Have the authors made all data underlying the findings in their manuscript fully available?

Reviewer #2: Yes

5. Is the manuscript presented in an intelligible fashion and written in standard English?

Reviewer #2: Yes

6. Review Comments to the Author

Reviewer #2: Authors have addressed the comments. I recommend to italicize the scientific name of the species in the references section.

7. PLOS authors have the option to publish the peer review history of their article (what does this mean?). If published, this will include your full peer review and any attached files.

Reviewer #2: No

---

## [Editor Report · Acceptance letter]

28 Sep 2020

PONE-D-19-32934R1 

DArTSeq SNP-based markers revealed high genetic diversity and structured population in Ethiopian cowpea [Vigna unguiculata (L.) Walp] germplasms 

Dear Dr. Yao:

I'm pleased to inform you that your manuscript has been deemed suitable for publication in PLOS ONE. Congratulations! Your manuscript is now with our production department. 

Kind regards, 

on behalf of

Dr. Tzen-Yuh Chiang 

Academic Editor

PLOS ONE